# CLASSIFIER-DRIVEN DIFFUSION MODEL AND PLUG-AND-PLAY OF WEAKLY-SUPERVISED LEARNING FOR CONDITIONAL GENERATION

## ABSTRACT

Can a diffusion model for conditional generation be trained as a classifier? We address this question with the Classifier-Driven Diffusion Model (CLDDM), which trains a diffusion model by minimizing a per-timestep cross-entropy loss under class-label supervision, while achieving high-quality class-conditional generation. In other words, CLDDM establishes a unified framework that demonstrates the equivalence between classification and generation. This equivalence enables new training strategies for conditional diffusion models. In particular, we show that "weakly supervised" generation can be realized by leveraging established classification objectives from weakly supervised learning. Experimental results on a toy dataset and image benchmarks demonstrate both quantitative and qualitative equivalence between CLDDM and standard diffusion models, and further confirm that CLDDM supports conditional generation under weak supervision, such as learning with noisy labels and learning from label proportions.

## 1 INTRODUCTION

A diffusion model (Ho et al., 2020; Nichol & Dhariwal, 2021; Dhariwal & Nichol, 2021; Song et al., 2021b; Karras et al., 2022) trains the noise predictor $\varepsilon_\theta$ (e.g., UNet (Ronneberger et al., 2015)) by minimizing a per-timestep noise prediction loss $\ell_{\mathrm{DM}}$ as shown by Figure 1 (a). Despite this simple objective, they have become state-of-the-art generative models across images. To control generated images, there are many studies on conditional diffusion models, such as text (Rombach et al., 2022; Podell et al., 2024; Sauer et al., 2025; Esser et al., 2024; Nichol et al., 2022; Saharia et al., 2022) and image-guided (Zhang et al., 2023; Wang et al., 2022; Cheng et al., 2024; Li et al., 2023c; Mou et al., 2024) generation.

Recent works have proposed the framework of how to use the conditional diffusion model for classification (Jeong et al., 2025; Li et al., 2023a; Clark & Jaini, 2023; He et al., 2024; Krojer et al., 2023) as shown by Figure 1 (b). The framework uses a trained diffusion model to compute the noise prediction loss for all classes on the input data. The input data are categorized into a class that minimizes the loss across all classes. The framework leverages generative diffusion models that clearly understand the structure of each class for the classification task. In other words, the framework flows: *train generation, then use for classification*.

By contrast, the reverse direction remains underexplored: *train classification, then use for generation*. To the best of our knowledge, within diffusion models and without generative objectives, no prior work has demonstrated direct conditional generation from purely classification training.

We answer this question with the **Classifier-Driven Diffusion Model (CLDDM)**, a framework that trains a per-timestep classifier $p_\theta$ by minimizing a per-timestep cross-entropy loss, as shown by Figure 1 (c) for conditional generation. Our framework leverages the classifier $p_\theta$ based on the conditional noise predictor $\varepsilon_\theta$. Consequently, we can train the noise predictor $\varepsilon_\theta$ indirectly for conditional generation through the training of classification for the classifier $p_\theta$. In addition, we provide equivalence between the cross-entropy loss in our framework and the noise prediction loss in the standard diffusion model, a principled explanation for why classification training can suffice for generation while using no generative objectives directly.

Figure 1: (a) Standard diffusion model trains the noise predictor $\varepsilon_\theta$ with the mean squared error (MSE) loss between predicted noise and added noise. (b) The conventional method uses the diffusion model for classification. The Classifier-Driven Diffusion Model trains the classifier $p_\theta$ based on the noise predictor $\varepsilon_\theta$ with a per-timestep cross-entropy (CE) loss for conditional generation with (c) supervised learning, and (d) weakly-supervised learning.

Moreover, the CLDDM provides a unified framework for weakly-supervised learning of the diffusion model as shown by Figure 1 (d). A generative model with weak supervision, such as noisy labels (Song et al., 2023; Algan & Ulusoy, 2021; Han et al., 2018; Patrini et al., 2017), partial labels (Cour et al., 2011; Lv et al., 2020; Feng et al., 2020), complementary labels (Ishida et al., 2017; Yu et al., 2018; Ishida et al., 2019), and label proportions (Zhang et al., 2022; Ardehaly & Culotta, 2017; Asanomi et al., 2023), is a long-standing challenge because we need complex training algorithms for each weak supervision. Therefore, there are few studies for diffusion models with weak supervision (Na et al., 2024; Hoang et al., 2024; Takahashi et al., 2025).

We introduce the **plug-and-play of weakly-supervised learning for conditional generation**. Because the CLDDM trains by classification with cross-entropy for generation, we leverage well-studied WSL losses based on cross-entropy to establish WSL for the diffusion model. In practice, this allows us to plug in forward correction (Patrini et al., 2017) for learning with noisy labels and proportion-based objectives (Ardehaly & Culotta, 2017) for learning from label proportions. This turns long-standing challenges in generative modeling into a simple recipe.

Through experiments on a toy dataset using 2D Gaussian mixtures and image data (MNIST (LeCun et al., 1998) and CIFAR10 (Krizhevsky, 2009)), we show that the CLDDM trained with classification objectives matches the conditional generative quality of the standard diffusion model. Moreover, for learning with noisy labels and learning from label proportions as weakly-supervised learning, the CLDDM trained with weakly-supervised losses enables conditional generation.

Our main contributions are as follows:

- We propose the Classifier-Driven Diffusion Model (CLDDM), a diffusion model trained with per-timestep cross-entropy for classification, but that nevertheless supports direct class-conditional generation.

- The CLDDM provides the plug-and-play of weakly-supervised learning that is a unified weakly-supervised learning recipe for conditional generation: existing classification objectives for weak supervision (e.g., noisy labels and label proportions) can be plugged in.

- We evaluate the CLDDM on 2D Gaussian mixtures and image data, demonstrating that quantitative and qualitative results are comparable to standard diffusion models on supervised learning, and enabling conditional generation with weakly-supervised learning.

## 2 RELATED WORK

**Discriminative Training for Generative Models.** Using a discriminative objective to learn a generative model has several approaches. A popular approach is the generative adversarial network (GAN) (Goodfellow et al., 2014; Arjovsky et al., 2017; Karras et al., 2019). GAN optimizes a generator via an adversarial discriminator. Another approach is density-ratio estimation that is formulated likewise to recover data distributions from discriminative objectives (Yadin et al., 2024; Lazarow et al., 2017; Grover & Ermon, 2018; Choi et al., 2021; 2022; Rhodes et al., 2020). For diffusion models, the classification diffusion model uses a discriminative objective for generative models via density estimation (Yadin et al., 2024). The models are trained by cross-entropy for timestep classification. Moreover, this model uses generative noise prediction loss, as in the standard diffusion model, during training.

Our proposed method has a different view compared with the previous approaches. The previous approaches mainly use the discriminative objective that discriminates between real and generated data for unconditional generation. Meanwhile, our proposed models are trained by a "classification" objective of multi-class classification for a conditional generation.

**Generative Diffusion Models for Classification.** Building the classifiers based on generatively trained diffusion models has been proposed. For example, zero-shot classification is performed using text-to-image diffusion models (Jeong et al., 2025; Li et al., 2023a; Clark & Jaini, 2023; He et al., 2024; Krojer et al., 2023), such as stable diffusion (Rombach et al., 2022). Moreover, the classifier from generative diffusion models also has adversarial robustness for the classification task (Chen et al., 2024). These methods follow "train generatively, use for classification." While we investigate the reverse: "train discriminatively, use for generation." Therefore, our work focuses on diffusion models trained only with classification objectives and their use as conditional generators.

**Weakly-supervised Learning for Diffusion Model** Weakly-supervised learning (WSL) offers generic objectives for incomplete supervision—noisy labels (Song et al., 2023; Algan & Ulusoy, 2021; Han et al., 2018; Patrini et al., 2017), partial labels (Cour et al., 2011; Lv et al., 2020; Feng et al., 2020), complementary labels (Ishida et al., 2017; Yu et al., 2018; Ishida et al., 2019), and label proportions (Zhang et al., 2022; Ardehaly & Culotta, 2017; Asanomi et al., 2023)-studied in a classification task. In diffusion models, weakly-supervised learning remains limited and often requires weak-label-specific designs and training algorithms (Na et al., 2024; Hoang et al., 2024; Takahashi et al., 2025). In our methods, by training diffusion models solely with classification objectives, we can plug in established WSL objectives from classification in a simple, unified recipe for conditional generation, without specific designs for generative WSL.

## 3 CLASSIFIER-DRIVEN DIFFUSION MODEL

In this section, we propose the Classifier-Driven Diffusion Model (CLDDM) that trains classification for conditional generation. The CLDDM uses a per-timestep classifier $p_\theta$ that takes a noise-added example and outputs class probabilities. The classifier is based on the noise predictor $\varepsilon_\theta$ of the standard diffusion model, and is trained with per-timestep cross-entropy. Interestingly, after the training of classification, we can obtain the noise predictor for conditional generation. Moreover, we introduce the plug-and-play of weakly-supervised learning (WSL) for conditional generation in the CLDDM. When the per-timestep classifier is trained by WSL objectives from classification instead of standard cross-entropy, surprisingly, we can establish the WSL for diffusion models.

### 3.1 PRELIMINARY: STANDARD DIFFUSION MODEL

We briefly introduce the standard diffusion model. The model consists of the diffusion and the reverse process based on a Gaussian distribution. We use a neural network $\varepsilon_\theta$, where $\theta$ is a parameter, that predicts Gaussian noise in sampling based on the reverse process. The training of the diffusion model applies per-timestep noise prediction loss, which is a mean squared error (MSE) loss between the added noise and the predicted noise of the neural network $\varepsilon_\theta$. Moreover, for this introduction, we set a labeled data example, real data $x_0$ and class label $y \in \{1, 2, \ldots, K\}$, where $K$ is the number of classes.

**Diffusion process.** Here is a discrete-time diffusion framework (Ho et al., 2020; Nichol & Dhariwal, 2021; Dhariwal & Nichol, 2021) with a diffusion (a.k.a. forward) process[1]. It progressively corrupts data and a learned reverse (denoising) process that inverts it. Given timesteps $t \in \{1, 2, \dots, T\}$ and a noise schedule $\{\beta_t\}_{t=1}^{T}$, we define $\alpha_t = 1 - \beta_t$ and $\bar{\alpha}_t = \prod_{s=1}^{t} \alpha_s$. The diffusion process and the closed form with respect to $x_0$ are

$$q(x_t \mid x_{t-1}) = \mathcal{N}\big(\sqrt{\alpha_t}\, x_{t-1},\, (1 - \alpha_t)\mathbf{I}\big), \qquad q(x_t \mid x_0) = \mathcal{N}\big(\sqrt{\bar{\alpha}_t}\, x_0,\, (1 - \bar{\alpha}_t)\mathbf{I}\big), \tag{1}$$

which equivalently yields the reparameterized form

$$x_t = \sqrt{\bar{\alpha}_t}\, x_0 + \sqrt{1 - \bar{\alpha}_t}\, \varepsilon, \qquad \varepsilon \sim \mathcal{N}(0, \mathbf{I}). \tag{2}$$

As $t$ increases, the signal-to-noise ratio decreases and $x_T$ approaches an isotropic Gaussian.

**Reverse process.** The forward chain implies a tractable posterior $q(x_{t-1} \mid x_t, x_0)$:

$$q(x_{t-1} \mid x_t, x_0) = \mathcal{N}\big(\tilde{\mu}_t(x_t, x_0),\, \beta_t \mathbf{I}\big), \quad \tilde{\mu}_t(x_t, x_0) = \frac{\sqrt{\bar{\alpha}_{t-1}}\beta_t}{1 - \bar{\alpha}_t}\, x_0 + \frac{\sqrt{\alpha_t}(1 - \bar{\alpha}_{t-1})}{1 - \bar{\alpha}_t}\, x_t, \tag{3}$$

We model the reverse process as a Gaussian Markov chain initialized at $r(x_T) = \mathcal{N}(0, \mathbf{I})$,

$$r_\theta(x_{0:T}) = r(x_T) \prod_{t=1}^{T} r_\theta(x_{t-1} \mid x_t), \qquad r_\theta(x_{t-1} \mid x_t) = \mathcal{N}\big(\mu_\theta(x_t, t),\, \beta_t\big). \tag{4}$$

Here, $\mu_\theta$ is a neural network with a parameter $\theta$.

In practice, the model $\mu_\theta$ is parametrized by the noise predictor $\varepsilon_\theta$. We use the time $t$- and class $y$-conditioned noise predictor $\varepsilon_\theta(x_t, t, y)$ for a conditional diffusion model. From Equation 2, we reconstruct $\hat{x}_0$ as

$$\hat{x}_0(x_t, t, y) = \frac{1}{\sqrt{\bar{\alpha}_t}}\Big(x_t - \sqrt{1 - \bar{\alpha}_t}\, \varepsilon_\theta(x_t, t, y)\Big), \tag{5}$$

and plug it into the posterior mean Equation 3 to obtain

$$\mu_\theta(x_t, t, y) = \tilde{\mu}_t\big(x_t, \hat{x}_0(x_t, t, y)\big). \tag{6}$$

For the generation of new data, we start the sampling step from the pure noise $x_T \sim \mathcal{N}(0, \mathbf{I})$, and then compute

$$x_{t-1} = \mu_\theta(x_t, t, y) + \sqrt{\beta_t}\, z, \qquad z \sim \mathcal{N}(0, \mathbf{I}). \tag{7}$$

Deterministic updates such as DDIM (Song et al., 2021a) can be obtained by setting the stochastic term to zero with an appropriate mean update, but we use Equation 7 as the default.

**Per-timestep noise prediction loss.** For training the noise predictor $\varepsilon_\theta$, the diffusion model uses the per-timestep noise prediction loss, which is the MSE loss between added noise and predicted noise. Given an example $x_0$, a label $y$, and a timestep $t$, the loss function is defined as following:

$$\ell_{\mathrm{DM}}(x_t, t, y) = \big\|\varepsilon_\theta(x_t, t, y) - \varepsilon\big\|^2, \qquad \text{with} \quad x_t = \sqrt{\bar{\alpha}_t}\, x_0 + \sqrt{1 - \bar{\alpha}_t}\, \varepsilon,\ \varepsilon \sim \mathcal{N}(0, \mathbf{I}). \tag{8}$$

## 3.2 CLASSIFICATION TRAINING WITH CROSS-ENTROPY FOR DIFFUSION MODEL

Based on the standard diffusion model, we construct the per-timestep classifier $p_\theta$ as shown by Figure 1 (c). We need to build this per-timestep classifier that consists of the noise predictor $\varepsilon_\theta$ because we finally must use the trained noise predictor to generate images. Consequently, we use the noise predictor $\varepsilon_\theta$ to calculate "negative" per-timestep noise prediction loss $-\ell_{\mathrm{DM}}(x_t, t, y)$ for all classes and apply the softmax function. We can obtain the classifier at each timestep:

$$p_\theta(y \mid x_t, t) = \frac{\exp\big(-\ell_{\mathrm{DM}}(x_t, t, y)\big)}{\sum_{\hat{y}=1}^{K} \exp\big(-\ell_{\mathrm{DM}}(x_t, t, \hat{y})\big)}, \tag{9}$$

---

[1]Our method can also be applied naturally for continuous-time diffusion models frameworks (Song et al., 2021b; Karras et al., 2022)

where the numerator is related to the noise prediction of the target class and the denominator to all classes. The classifier defines lower noise prediction loss at timestep $t$ as indicating greater compatibility with target class label $y$ through class posterior probability. We can obtain the trained noise predictor $\varepsilon_\theta$ through the training with the proposed loss function later.

We *purely class-discriminatively* train the per-timestep classifier. Given an example $x_0$ and a class label $y^\star$, we sample a single timestep $t$ and a random noise $\varepsilon$, construct $x_t = \sqrt{\bar{\alpha}_t}x_0 + \sqrt{1 - \bar{\alpha}_t}\,\varepsilon$, evaluate the class probabilities via Equation 9, and minimize the standard cross-entropy loss:

$$\mathcal{L}_{\mathrm{CLDDM}}(x_t, t, y^\star) = \underbrace{-\sum_{y=1}^{K} \mathbf{1}[y = y^\star]\, \log\, p_\theta(y \mid x_t, t)}_{\mathcal{L}_{\mathrm{ce}}} \underbrace{-\lambda_{\mathrm{reg}} \log \sum_{y=1}^{K} \exp\big(-\ell_{\mathrm{DM}}(x_t, t, y)\big)}_{\mathcal{L}_{\mathrm{reg}}}.$$

$$(10)$$

where the first term $\mathcal{L}_{\mathrm{ce}}$ is equal to the cross-entropy loss of the classification task, and the second term $\mathcal{L}_{\mathrm{reg}}$ is the regularization term that is introduced to prevent divergence of the denominator in Equation 9. $\lambda_{\mathrm{reg}}$ is a hyperparameter to balance two terms. The loss function uses no generative objective directly; the model is optimized via the cross-entropy in Equation 10 using logits given by $-\ell_{\mathrm{DM}}(x_t, t, y)$.

The per-timestep classifier $p_\theta$ trained by $\mathcal{L}_{\mathrm{CLDDM}}$ can build the noise predictor $\varepsilon_\theta$ for conditional generation without generative objectives. Crucially, when $\lambda_{\mathrm{reg}} = 1$, the regularization term exactly cancels the normalization term arising in the cross-entropy, so that $\mathcal{L}_{\mathrm{CLDDM}}$ coincides with the standard noise prediction loss $\ell_{\mathrm{DM}}$. Therefore, classification-only training provably recovers generative training at the objective level. We refer the reader to Appendix A for a proof of this equivalence.

### 3.3 PLUG-AND-PLAY OF WEAKLY-SUPERVISED LEARNING FOR CONDITIONAL GENERATION

The proposed CLDDM provides a very simple training strategy for a conditional diffusion model. The CLDDM uses only cross-entropy and a simple regularization term. The cross-entropy style loss functions are very popular with classification tasks. Fortunately, a lot of methods for weakly-supervised learning (WSL) leverage the cross-entropy style (Patrini et al., 2017; Ardehaly & Culotta, 2017; Asanomi et al., 2023; Yu et al., 2018; Feng et al., 2020). Therefore, we can easily apply these WSL loss functions to the CLDDM.

Consequently, we introduce plug-and-play of weakly-supervised learning for the conditional diffusion model in the CLDDM. The plugin only replaces a standard cross-entropy loss $\mathcal{L}_{\mathrm{ce}}$ in Equation 10 with a weakly-supervised loss following cross-entropy style as shown by Figure 1 (d). The conventional method for WSL of diffusion models needs to consider complex training algorithms or architectures for each WSL task. By contrast, our plug-and-play provides a simple strategy for WSL of diffusion models since the method only uses WSL losses from classification. This property opens the door to WSL in diffusion models easily.

In this study, we instantiate two cases: Learning with Noisy Labels (LNL) (Song et al., 2023; Algan & Ulusoy, 2021; Han et al., 2018; Patrini et al., 2017) and Learning from Label Proportions (LLP) (Zhang et al., 2022; Ardehaly & Culotta, 2017; Asanomi et al., 2023). Because these cases have loss functions followed by the cross-entropy style, we select them.

**Learning with Noisy Labels (LNL).** LNL means the training with the dataset that includes mislabeling examples (a.k.a., noisy labels). Noisy labels cause misclassification of a classifier and mismatched conditional generation of a generator. As a remedy for noisy labels for classification, the forward correction framework has been proposed by (Patrini et al., 2017).

We leverage the forward correction framework for LNL of CLDDM. The framework assumes mislabeling with a transition matrix $T \in \mathbb{R}^{K \times K}$ whose entries $T_{y, \tilde{y}} = \Pr(\tilde{y} \mid y)$ give the probability that a true class $y$ is observed as $\tilde{y}$. For a given timestep $t$, the posterior of per-timestep classifiers over true classes is $p_\theta(\cdot \mid x_t, t)$; the induced distribution over observed labels is

$$\tilde{p}_\theta(\tilde{y} \mid x_t, t) = \big[T^\top p_\theta(\cdot \mid x_t, t)\big]_{\tilde{y}} = \sum_{y=1}^{K} T_{y, \tilde{y}}\, p_\theta(y \mid x_t, t). \qquad (11)$$

Replacing the standard cross-entropy loss with its forward-corrected counterpart yields the noisy label loss

$$\mathcal{L}_{\text{noisy}}(x_t, t, \tilde{y}) \;=\; -\log \sum_{y=1}^{K} T_{y,\tilde{y}}\, p_\theta(y \mid x_t, t). \tag{12}$$

Compared to the standard cross-entropy loss, the only change is that the cross-entropy is evaluated on the label-noise mapped distribution $T^\top p_\theta(\cdot \mid x_t, t)$. We use $\mathcal{L}_{\text{noisy}} - \lambda_{\text{reg}}\mathcal{L}_{\text{reg}}$ as loss functions for the training of the CLDDM on LNL.

**Learning from Label Proportions (LLP).** LLP is a learning method on a set of examples, given a label proportion for the examples. In some situations, example-level labelings cost a lot, and there are many situations in which example-level labeling cannot be accessed for privacy reasons. In such a situation, we only know the label proportions given to a set of examples (bag). For the LLP with the classification task, the proportion loss (Ardehaly & Culotta, 2017) is a standard approach, and many LLP methods are based on it. We apply the proportion loss to CLDDM for LLP with the generation task.

Given a bag $\{x_{0,n}\}_{n=1}^B$ with a vector of label proportion $y_{\text{prop}}$, where $x_{0,n}$ is the $n$-th examples and $B$ is the number of examples (bag size). The label proportion $y_{\text{prop}}$ is a $K$ elements vector $(y_{\text{prop}}^1, y_{\text{prop}}^2, \dots, y_{\text{prop}}^K)$ and each element means the proportion of labels corresponding to examples in a bag, where $\sum_{i=1}^K y_{\text{prop}}^i = 1$. The proportion loss is a bag-level cross-entropy between the ground-truth of the label proportions and the predicted proportions, which is the average of the classifier's output class probabilities in a bag. At our per-timestep classifiers, the proportion loss is defined as:

$$\mathcal{L}_{\text{prop}}(\{x_{t,n}\}_{n=1}^B, t, y_{\text{prop}}) = -\sum_{m=1}^K y_{\text{prop}}^m \log\left(\frac{1}{B}\sum_{n=1}^B p_\theta(y = m \mid x_{t,n}, t)\right), \tag{13}$$

where $x_{t,n}$ is the $n$-th example at timestep $t$ following Equation 2. The CLDDM uses $\mathcal{L}_{\text{prop}} - \lambda_{\text{reg}}\mathcal{L}_{\text{reg}}$ as loss functions on the training of LLP.

## 4 EXPERIMENTS

In our experiments, we evaluated how similar the classifier-driven diffusion model (CLDDM) is to the standard diffusion model (DM) because the CLDDM of the training task is different from the DM, but the target task is the same as a class-conditional generation. Therefore, we compared the CLDDM with the DM on quantitative and qualitative results for *supervised* learning (SL). As evaluation datasets, we used conditional generation tasks of 2D Gaussian mixtures and image data (MNIST and CIFAR10). Moreover, we evaluated the plug-and-play of the CLDDM for *weakly-supervised* learning (WSL) at learning with noisy labels (LNL), and learning from label proportions (LLP).

### 4.1 EXPERIMENTAL SETTINGS

**Datasets.** We use (i) **2D Gaussian mixtures**: $K{=}10$ classes with means $\{\mu_k\}$ uniformly on the unit circle ($R{=}1$), isotropic components $x \mid y{=}k \sim \mathcal{N}(\mu_k, \sigma^2 I_2)$, and two difficulty levels $\sigma \in \{0.05, 0.1\}$; 1,000 samples are drawn from the same generative rule for train/eval. For WSL on 2D, we use $\sigma{=}0.05$. (ii) **Images**: MNIST and CIFAR10; only the official training splits are used for model training and to compute reference statistics. For image-domain WSL, we use MNIST. **WSL settings**: LNL uses a symmetric transition matrix and noise flip with noise rate $\eta{=}0.2$ (Han et al., 2018; Patrini et al., 2017); LLP uses bag size 32 with Dirichlet-sampled proportions and bags formed without replacement (Zhang et al., 2022). Please see the details of Appendix B

**Metrics.** **2D**: we report Wasserstein-2 distance ($W_2$) between real and generated distributions in two views—unconditional (all classes mixed) and class-wise (per-class, then averaged); 1,000 generated samples are used and results are averaged over five times. **Images**: we report FID (Heusel et al., 2017)/Precision/Recall (Kynkäänniemi et al., 2019) and CW-FID/CW-Precision/CW-Recall and CAS (classification accuracy score on generated data) (Ravuri & Vinyals, 2019), where the class-wise (CW) evaluates the metric separately for each class and averages them. Following prior

Table 1: Quantitative results at the 2D Gaussian mixtures for SL. Show $W_2$ ($\times 10^{-2}$) results. "CW" means class-wise results. All results are averaged over five times

| Dataset | Method | $W_2 \downarrow$ | CW-$W_2 \downarrow$ |
|---|---|---|---|
| $\sigma = 0.05$ | DM | 2.64 | 2.60 |
| | CLDDM | 2.64 | 2.61 |
| $\sigma = 0.1$ | DM | 5.76 | 5.73 |
| | CLDDM | 5.76 | 5.73 |

Table 2: The results of WSL on the 2D Gaussian mixtures with $\sigma = 0.05$. "CLDDM w/ " uses loss functions for WSL. See the caption of Table 1 for details.

| Task | Method | $W_2 \downarrow$ | CW-$W_2 \downarrow$ |
|---|---|---|---|
| LNL | DM | 13.0 | 120 |
| | CLDDM | 13.0 | 120 |
| | CLDDM w/ $\mathcal{L}_{\text{noisy}}$ | 2.71 | 2.67 |
| LLP | CLDDM w/ $\mathcal{L}_{\text{prop}}$ | 3.01 | 2.96 |

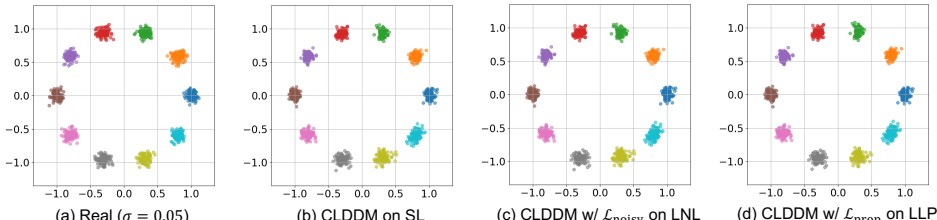

(a) Real ($\sigma = 0.05$)    (b) CLDDM on SL    (c) CLDDM w/ $\mathcal{L}_{\text{noisy}}$ on LNL    (d) CLDDM w/ $\mathcal{L}_{\text{prop}}$ on LLP

Figure 2: Real data and generated data at 2D Gaussian mixtures with $\sigma = 0.05$. Each color corresponds to a class.

work, MNIST metrics use a randomly initialized VGG-16 as the feature extractor, as suggested by previous work (Naeem et al., 2020); CIFAR10 uses Inception-v3. We generate 50,000 images for evaluation and **omit FID on MNIST** as it is not suitable for this dataset, as suggested by (Na et al., 2024). A detailed description for the metrics is at Appendix C.

**Implementation details.** **2D**: we use a simple MLP (128–128, SiLU) that serves as the time- and class-conditional noise predictor; linear $\beta$ schedule with $\beta \in [10^{-5}, 10^{-2}]$, $T=50$ steps. For the training, we set an iteration of 10k, a batch size of 512 (LLP: 8), and the Adam optimizer with a learning rate of $10^{-3}$. The regularization weight $\lambda_{\text{reg}}$ is $\{1.0, 0.6, 0.5\}$ for $\{$SL,LNL,LLP$\}$. **Images**: we use a class-conditional U-Net, following the previous work (Song et al., 2021b), with a *multi-head* top ($K$ parallel outputs) (Chen et al., 2024) that enables per-timestep prediction for all classes in one forward pass. The optimizer is Adam with a learning rate of $10^{-4}$. The other settings (e.g., backbones/schedules/samplers) are based on EDM (Karras et al., 2022). Training iterations are 500k (MNIST) and 1M (CIFAR10), batch 64 (LLP: 8); $\lambda_{\text{reg}}$ is $\{1.0, 0.3, 0.01\}$ for $\{$SL,LNL,LLP$\}$. Please see the details of Appendix D

### 4.2 Experimental Results of 2D Gaussian Mixtures

Table 1 shows quantitative results using unconditional $W_2$ and class-wise $W_2$ on 2D Gaussian mixtures with supervised learning. Across both difficulty levels, CLDDM achieved essentially the same scores as the generatively trained diffusion model (DM). This indicates that a diffusion model trained solely with classification objectives can attain the same conditional generative quality as a model trained with the standard noise prediction loss.

The results of weakly-supervised learning are summarized in Table 2. With LNL, a naive training of DM or CLDDM degraded performance because some of the class labels were mislabeled in the dataset. Whereas CLDDM that uses $\mathcal{L}_{\text{noisy}}$ (CLDDM w/ $\mathcal{L}_{\text{noisy}}$) instead of $\mathcal{L}_{\text{ce}}$ significantly improved both unconditional and class-wise $W_2$, nearly matching the supervised learning as shown by Table 1. In LLP, CLDDM w/ $\mathcal{L}_{\text{prop}}$ also provided similar quantitative results to supervised learning.

Figure 2 shows qualitative comparison of generated data with real data on 2D Gaussian mixtures. On supervised learning, the CLDDM reproduced the geometry of the ground-truth distribution faithfully: clusters are placed at the correct locations with the correct spread, and no mode collapse or

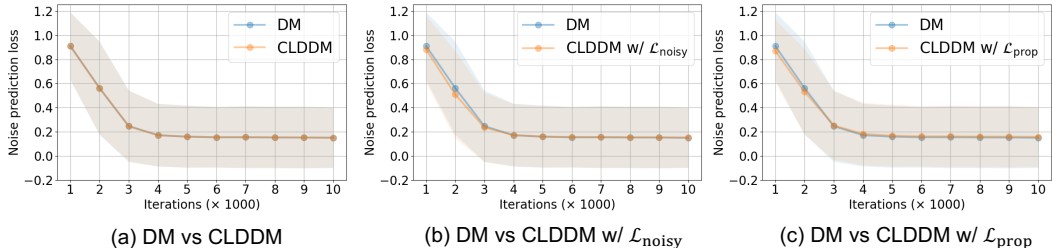

(a) DM vs CLDDM                (b) DM vs CLDDM w/ $\mathcal{L}_{\text{noisy}}$                (c) DM vs CLDDM w/ $\mathcal{L}_{\text{prop}}$

Figure 3: Comparison of noise prediction loss (Equation 8) curves with the standard diffusion model (DM) of SL vs CLDDMs of SL and WSL (LNL and LLP) on the dataset of 2D Gaussian mixtures with $\sigma = 0.05$. These loss curves are averaged over five times and all timesteps, and the width of the shading indicates the standard deviation. The monitored loss curves of DM (blue line) and CLDDMs (orange lines) are almost perfectly overlapped across SL, LNL, and LLP settings, empirically supporting our equivalence analysis.

Table 3: Quantitative results of *supervised* learning for class-conditional image generation on the MNIST, CIFAR10 datasets. "Pre." and "Rec." indicate Precision and Recall.

| Dataset | Method | uncond | | | cond | | | |
|---|---|---|---|---|---|---|---|---|
| | | Pre.↑ | Rec.↑ | FID ↓ | CAS↑ | CW-Pre.↑ | CW-Rec.↑ | CW-FID↓ |
| MNIST | DM | 0.822 | 0.852 | N/A | 0.907 | 0.787 | 0.865 | N/A |
| | CLDDM | 0.818 | 0.856 | N/A | 0.902 | 0.786 | 0.866 | N/A |
| CIFAR10 | DM | 0.755 | 0.715 | 4.91 | 0.366 | 0.599 | 0.697 | 37.4 |
| | CLDDM | 0.758 | 0.726 | 4.86 | 0.368 | 0.596 | 0.700 | 37.1 |

unintended merging is observed. Under weakly-supervised learning, both CLDDM w/ $\mathcal{L}_{\text{noisy}}$ and CLDDM w/ $\mathcal{L}_{\text{prop}}$ maintained class separation and variance, suggesting that WSL objectives can be applied without distorting the conditional structure.

Figure 3 compares the original noise prediction loss (Equation 8) between the DM on supervised learning (SL) and CLDDMs with three learning settings (SL, LNL, and LLP). Monitored during the training, the loss curves of CLDDMs of all settings and the DM were almost perfectly overlapped. This empirical alignment supports our theoretical analysis, confirming that classification training drives essentially the same optimization dynamics as generative training.

From these results, we can conclude that the CLDDM can train the conditional generation through a classification objective, and plug-and-play of weakly-supervised learning is effective, although a simple strategy.

### 4.3 EXPERIMENTAL RESULTS OF IMAGE GENERATION

Table 3 shows unconditional metrics (FID, Precision, Recall) and conditional metrics (CW-FID, CW-Precision, CW-Recall, CAS) for MNIST and CIFAR10 on supervised learning. Across all metrics, the CLDDM achieved essentially the same scores as the standard diffusion model (DM). In particular, the differences in FID and CW-FID were within the margin of error, showing that a diffusion model trained only on the classification objective achieved conditional generative performance on par with the standard generative training.

Figures 4 (a), (b), and (c) show qualitative comparisons of real and generated images in MNIST on SL. We confirmed that the standard diffusion model could generate realistic images corresponding to the class labels accurately. The CLDDM also produced visually faithful digits and objects across classes without mode collapse or anomalous samples, confirming that high-quality image generation could be achieved using purely classification training.

Table 4 shows the quantitative results for LNL and LLP on MNIST. Under LNL, we confirmed that a naive training of the DM/CLDDM weakened class–image alignment due to CAS degradation.

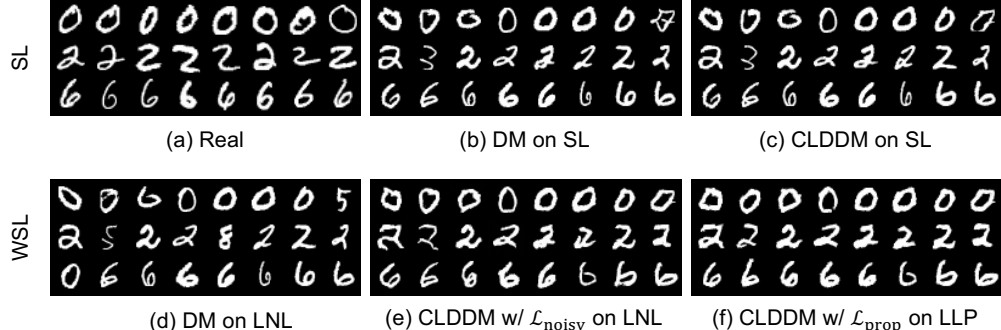

Figure 4: Real images and generated images in MNIST on supervised learning (SL) and weakly-supervised learning (WSL). Each row corresponds to the classes of "0", "2", and "6", respectively.

Table 4: Quantitative results of *weakly-supervised* learning on MNIST for LNL and LLP. "CLDDM w/ " uses loss functions for WSL. The results of supervised learning (SL) are shown as a reference.

| Task | Method | uncond | | cond | | |
|------|--------|--------|--------|--------|--------|--------|
| | | Pre.↑ | Rec.↑ | CAS↑ | CW-Pre.↑ | CW-Rec.↑ |
| SL | DM | 0.822 | 0.852 | 0.907 | 0.787 | 0.865 |
| LNL | DM | 0.822 | 0.858 | 0.660 | 0.678 | 0.876 |
| | CLDDM | 0.826 | 0.857 | 0.660 | 0.676 | 0.874 |
| | CLDDM w/ $\mathcal{L}_{\text{noisy}}$ | 0.753 | 0.847 | 0.985 | 0.692 | 0.828 |
| LLP | CLDDM w/ $\mathcal{L}_{\text{prop}}$ | 0.739 | 0.857 | 0.952 | 0.667 | 0.860 |

Figure 4 (d) shows the generated images from the DM under LNL that are sometimes misaligned with class labels (e.g., a misalignment of "0"→ "5" occurs at an image in the upper right corner).

Whereas, replacing cross-entropy with the noisy-label loss in the CLDDM (CLDDM w/ $\mathcal{L}_{\text{noisy}}$) restored conditional generation and yielded CAS that exceeded a naive training of LNL and achieved a CAS comparable to that of the DM under SL in Table 4. At the same time, unconditional and class-wise Precision/Recall were lower, reflecting a fidelity–consistency trade-off due to the shift in optimization target under weakly-supervised learning. Moreover, Figure 4 (e) shows the generated images from the CLDDM w/ $\mathcal{L}_{\text{noisy}}$ on LNL. These images remained class-consistent with clear structures and reasonable within-class variation. From these results, on LNL, the plug-and-play of WSL is effective for conditional image generation.

Under LLP, applying the proportion loss to CLDDM (CLDDM w/ $\mathcal{L}_{\text{prop}}$) showed the same pattern: strong CAS with comparable or slightly reduced fidelity in Table 4. Figure 4 (f) presents the generated images by CLDDM w/ $\mathcal{L}_{\text{prop}}$ on LLP. These images follow the intended class while retaining diversity; slight smoothing relative to full supervision is sometimes visible, consistent with the same trade-off. Therefore, under LLP, plug-and-play of WSL in the CLDDM also achieved accurate conditional generation with only a change to the loss functions.

## 5 CONCLUSION

We proposed the Classifier-Driven Diffusion Model (CLDDM), a framework that trains a diffusion model with purely classification objectives and yet enables conditional generation. Moreover, we introduced plug-and-play of weakly-supervised learning (WSL) that leverages WSL objectives from classification in the CLDDM. Using 2D Gaussian mixtures and image datasets, we confirmed that the CLDDM is able to perform conditional generation comparable to the standard diffusion model for supervised learning. We also demonstrated that the plug-and-play of WSL can generate images that correctly correspond to class labels. Future works will focus on the other WSL settings (e.g., complementary labels, partial labels, etc.), and evaluation of high-resolution images.

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

## A  PROOF OF EQUIVALENCE OF CLDDM AND DM

Here, we prove equivalence of $\mathcal{L}_{\text{CLDDM}}$ (Equation 10) and $\ell_{\text{DM}}$ (Equation 8) when $\lambda_{\text{reg}} = 1$:

$$\mathcal{L}_{\text{CLDDM}}(x_t, t, y^\star)$$

$$= -\sum_{y=1}^{K} \mathbf{1}[y = y^\star] \, \log p_\theta(y \mid x_t, t) - \log \sum_{y=1}^{K} \exp\big(-\ell_{\text{DM}}(x_t, t, y)\big)$$

$$= -\log p_\theta(y^\star \mid x_t, t) - \log \sum_{y=1}^{K} \exp\big(-\ell_{\text{DM}}(x_t, t, y)\big)$$

$$= -\log \frac{\exp\big(-\ell_{\text{DM}}(x_t, t, y^\star)\big)}{\sum_{y=1}^{K} \exp\big(-\ell_{\text{DM}}(x_t, t, y)\big)} - \log \sum_{y=1}^{K} \exp\big(-\ell_{\text{DM}}(x_t, t, y)\big)$$

$$= -\log \exp\big(-\ell_{\text{DM}}(x_t, t, y^\star)\big) + \log \sum_{y=1}^{K} \exp\big(-\ell_{\text{DM}}(x_t, t, y)\big) - \log \sum_{y=1}^{K} \exp\big(-\ell_{\text{DM}}(x_t, t, y)\big)$$

$$= \ell_{\text{DM}}(x_t, t, y^\star).$$

## B  SETTINGS OF DATASETS

### B.1  2D GAUSSIAN MIXTURES

Following prior work, we used class-conditional 2D Gaussian mixtures (Bang & Shim, 2018; Liu et al., 2020; Li et al., 2023b; Gushchin et al., 2023). We set the number of classes to $K{=}10$ and place the class means $\{\mu_k\}$ uniformly on a circle of radius $R{=}1$. Each class distribution is isotropic, $x \mid y{=}k \sim \mathcal{N}(\mu_k, \sigma^2 I_2)$. We control difficulty solely via the class standard deviation $\sigma$ and evaluate two levels: Easy with $\sigma{=}0.05$ and Medium with $\sigma{=}0.1$. Both the training and evaluation samples are from the same generative rule (same underlying population), and we sample 1k data points.

### B.2  IMAGE DATA

We used MNIST ($28{\times}28$, grayscale) (LeCun et al., 1998) and CIFAR10 ($32{\times}32$, color) (Krizhevsky, 2009) to evaluate conditional image generation. In both cases we used only the official training split to compute reference statistics for evaluation and to train models. So, we used 60k samples on MNIST and 50k samples on CIFAR10, where the number of classes is $K = 10$ on both datasets and the class proportions are uniform within the datasets. In weakly-supervised learning, we used MNIST, and the weakly-supervised settings are the same as 2D Gaussian mixtures.

### B.3  LEARNING WITH NOISY LABEL

In LNL, we used 2D Gaussian mixtures with $\sigma{=}0.05$. Referring to the previous studies (Han et al., 2018; Patrini et al., 2017), we used class-conditional label noise via a transition matrix $T$ with diagonal $1 - \eta$ and off-diagonals $\eta/(K - 1)$, using $\eta = 0.2$ (i.e., the mislabeling occurs with a probability of 0.2). We applied the transition matrix for the original dataset to make a dataset with noisy labels.

### B.4  LEARNING FROM LABEL PROPORTIONS

In LLP, the bag size is 32, and the sampling label proportion follows the Dirichlet distribution, which is a popular setting in LLP for a classification task (Zhang et al., 2022). Using these LLP settings, we made the bags from the original dataset without replacement.

## C  SETTING OF EVALUATION METRICS

### C.1  2D GAUSSIAN MIXTURES

We used two types of Wasserstein-2 ($W_2$) distance between real and generated data distributions following prior work: (i) Unconditional $W_2$ between the full empirical distributions (all classes mixed), and (ii) Class-wise $W_2$, obtained by computing $W_2$ per class and averaging over classes. The reference distributions are formed by large i.i.d. samples from the ground-truth generative rule, matched in size to generated samples. We generated 1k samples to ensure each sample had the same number of classes for evaluation, and all results are averaged over five times at different seeds.

### C.2  IMAGE DATA

We evaluated conditional generative models from multiple perspectives using three unconditional metrics—Fréchet Inception Distance (FID) (Heusel et al., 2017), Precision, and Recall (Kynkäänniemi et al., 2019), and four conditional metrics, namely CW-FID, CW-Precision, CW-Recall, and Classification Accuracy Score (CAS) (Ravuri & Vinyals, 2019), where Class-Wise (CW) means computing each metric separately for each class and then averaging across classes. The FID evaluates distribution distances between real and generated data in feature spaces. We did not evaluate FID for MNIST because this metric is not suitable for the evaluation of MNIST, as noted in (Na et al., 2024). The Precision and Recall are related to the fidelity and diversity of a distribution of generated images. As feature extractors for FID, Precision/Recall, we used a randomly initialized VGG-16 as suggested by (Naeem et al., 2020) for MNIST and Inception-v3 with ImageNet pretraining for CIFAR10. The CAS is the accuracy of the generated images, which is calculated by a classifier trained using real data. In all metrics, We generated 50k samples to ensure each sample had the same number of classes for evaluation.

## D  IMPLEMENTATION DETAILS

### D.1  2D GAUSSIAN MIXTURES.

We used a simple MLP with two hidden layers (128–128, SiLU) as the time- and class-conditional noise predictor $\varepsilon_\theta$. Time is provided as a normalized scalar embedding; classes use a learnable embedding, concatenated with $x_t$ and $t$ before the MLP. We adopt a linear $\beta$ schedule with $\beta \in [10^{-5}, 10^{-2}]$ and $T=50$ diffusion steps; standard DDPM coefficients (e.g., $\bar{\alpha}_t$) are used. The training iterations are 100k, and the batch size is 512, except for LLP, which is 8. We use the Adam optimizer with a learning rate of $10^{-3}$T. The parameters $\lambda_{\mathrm{reg}}$ for the regularization term are 1.0, 0.6, and 0.5 for supervised learning, LNL, and LLP, respectively.

### D.2  IMAGE DATA

The noise predictor is a class-conditional UNet (Ronneberger et al., 2015) following Elucidated Diffusion Models (EDM) (Karras et al., 2022). The optimizer is Adam with a learning rate of $10^{-4}$. We adhere to the EDM paper's recommended settings for noise distributions, preconditioning, and related scalings, and NFE is 35. For computational efficiency, as suggested by (Chen et al., 2024), we employ a multi-head design with $K$ parallel output heads on top of a shared trunk, so that for each timestep $t$ the model predicts the noises for all classes $K$ in a single forward pass. All comparisons use identical backbones and schedules across methods. The training iterations are 500k for MNIST and 1M for CIFAR10. The batch size is 64 except for LLP, and for LLP, it is 8. The parameters $\lambda_{\mathrm{reg}}$ for the regularization term are 1.0, 0.3, and 0.01 for supervised learning, LNL, and LLP, respectively.

### D.3  CODE AVAILABILITY

If accepted, we will release the code to facilitate reproducibility and further research.

## E  GENERATED DATA

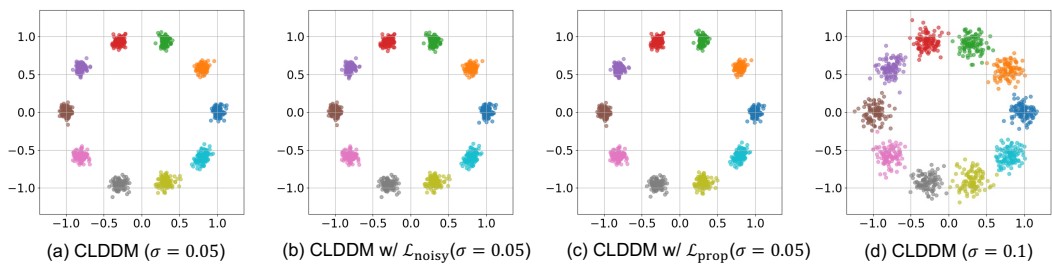

(a) CLDDM ($\sigma = 0.05$)   (b) CLDDM w/ $\mathcal{L}_{\text{noisy}}(\sigma = 0.05)$   (c) CLDDM w/ $\mathcal{L}_{\text{prop}}(\sigma = 0.05)$   (d) CLDDM ($\sigma = 0.1$)

Figure 5: Generated data at 2D Gaussian mixtures. Each color corresponds to a class.

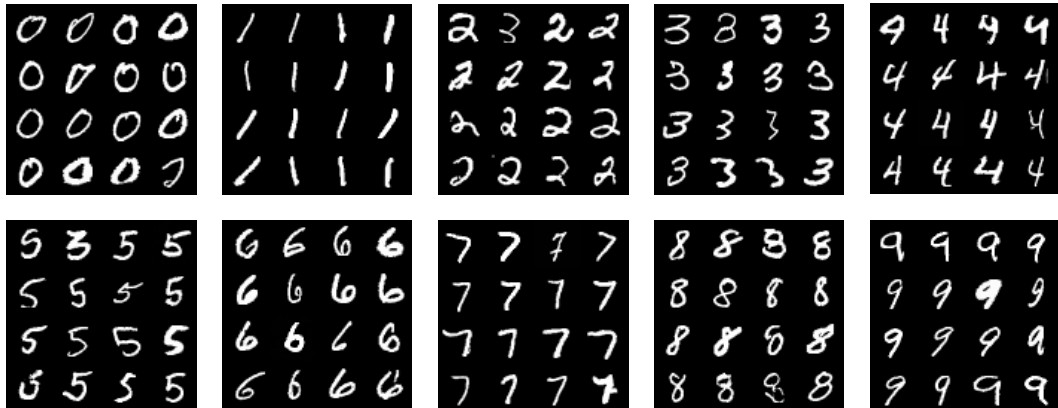

Figure 6: Generated images by CLDDM on supervised learning of MNIST. Classes from 0 to 9 are arranged in sequence from the top left

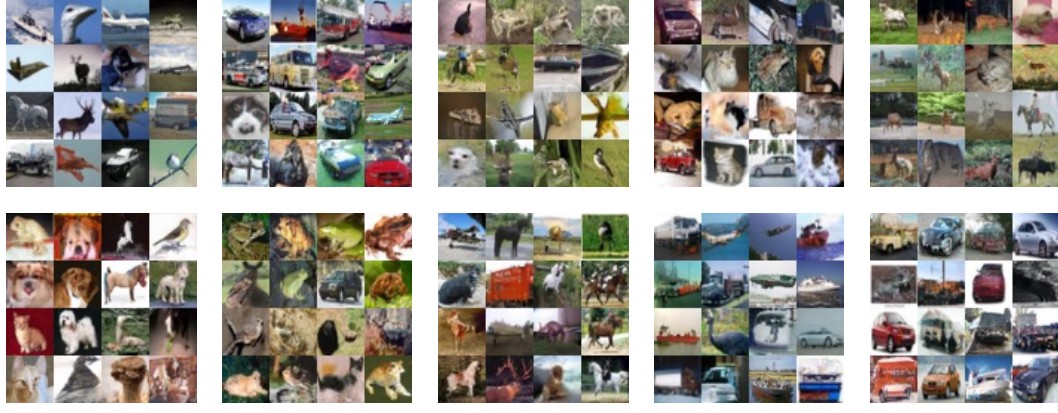

Figure 7: Generated images by CLDDM on supervised learning of CIFAR10. Classes are arranged in the order of "airplane", "automobile", "bird", "cat", "deer", "dog", "frog", "horse", "ship", and "truck" from top-left

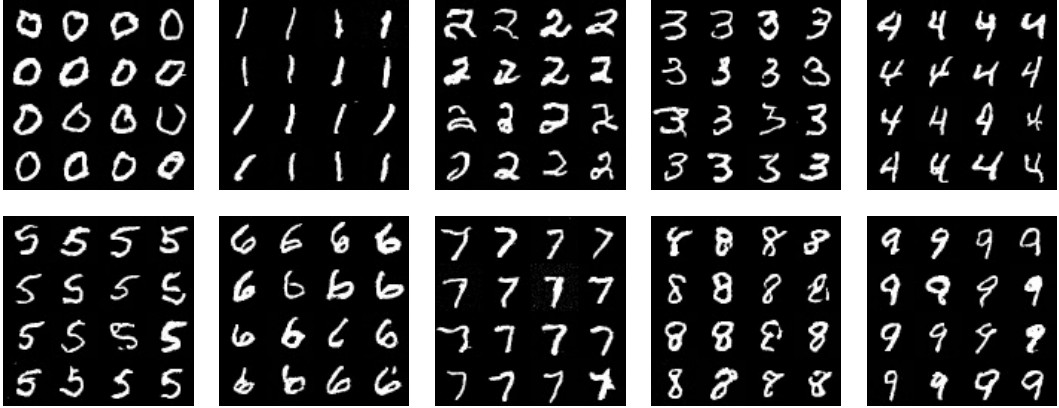

Figure 8: Generated images by CLDDM w/ $\mathcal{L}_{\text{noisy}}$ on learning with noisy labels of MNIST.

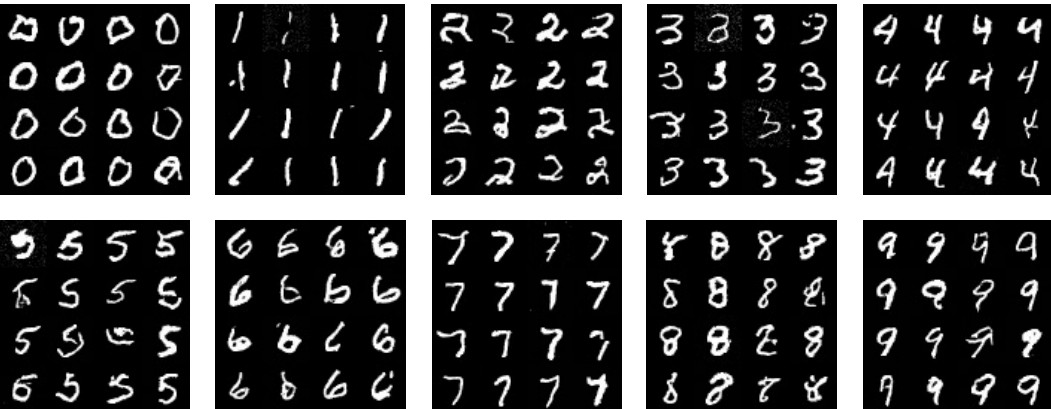

Figure 9: Generated images by CLDDM w/ $\mathcal{L}_{\text{prop}}$ on learning from label proportions of MNIST.

