# OpenReview forum: "Classifier-Driven Diffusion Model and Plug-and-Play of Weakly-Supervised Learning for Conditional Generation"
_ICLR.cc/2026/Conference — ICLR 2026 Conference Withdrawn Submission_

### Official Review · Reviewer_gBqt · 2025-10-20

**Soundness:** 3
**Presentation:** 3
**Contribution:** 3
**Rating:** 4
**Confidence:** 4

**Summary:**

The paper presents an innovative approach to diffusion modeling by first training a classifier and then leveraging it for generation. This is achieved by minimizing a per-timestep cross-entropy loss under class-label supervision during training. Experiments are conducted on the CIFAR-10 and 2D Gaussian mixture datasets to validate the method.

**Strengths:**

* The author presents an innovative investigation into how generative models can be utilized for classification tasks. In contrast, this work explores the reverse direction: training a classifier first, then using it for generation.

* The author demonstrates an equivalence between the cross-entropy loss used in their framework and the noise prediction loss employed in standard diffusion models.

**Weaknesses:**

* The motivation appears relatively weak. In line 42, the author states, "In other words, the framework flows: train generation, then use for classification." This aligns with intuition, as generation tasks (taking y as input to produce x ) are inherently more complex than classification tasks (taking x as input to predict y ). Therefore, it is reasonable to expect that generative models can support classification. However, the author aims to explore the reverse direction—"train classification, then use for generation"—which seems less intuitive and potentially more challenging.

* A notable limitation of the proposed "train classification, then use for generation" methodology is the requirement to specify class labels during cross-entropy training. This constraint may hinder the model's ability to perform zero-shot conditioned generation.

* The research evaluates the proposed methodology using CIFAR-10 and 2D Gaussian mixture datasets. However, I am concerned that the "train classification, then use for generation" approach may constrain the diversity and zero-shot capabilities of the diffusion model. Given the simplicity of the chosen datasets, the experiments may not sufficiently demonstrate whether generation diversity is preserved. While large-scale experiments are not strictly necessary, it would be valuable to include additional experiments that convincingly show CLDDM does not compromise diversity or generalization capacity.


* The results presented in Tables 3 and 4 do not provide compelling evidence that CLDDM achieves superior performance compared to standard diffusion models.

**Questions:**

* During cross-entropy training, in addition to computing the logit for the target class (used in the numerator), it is also necessary to calculate logits for all remaining classes to form the denominator. This raises the question: does this approach incur significantly more computational overhead compared to standard diffusion models, which typically require computation only for the corresponding class?

---

### Official Review · Reviewer_CrGb · 2025-10-27

**Soundness:** 3
**Presentation:** 3
**Contribution:** 1
**Rating:** 2
**Confidence:** 5

**Summary:**

The paper introduces Classifier-Driven Diffusion Models (CLDDM). This framework trains a conditional diffusion model purely with per-timestep classification objectives (cross-entropy plus a simple regularizer) yet still enables direct class-conditional generation. The key technical move is to define a per-timestep classifier using logits derived from the standard noise-prediction loss and to train with cross-entropy; when the regularization weight = 1, the objective is provably equivalent to the standard diffusion loss (Appendix A). The same construction yields a plug-and-play route to weakly-supervised learning (WSL) for diffusion models by swapping the cross-entropy with established WSL losses (e.g., forward correction for noisy labels and proportion loss for label proportions). Experiments on 2D Gaussian mixtures, MNIST, and CIFAR-10 show that CLDDM matches standard diffusion models on supervised training and enables conditional generation under WSL settings.

**Strengths:**

- Originality: The attempt to train a diffusion model with a class-discriminative loss is new.

- Scope: This paper covers supervised conditional learning as well as several semi-supervised settings.

**Weaknesses:**

- Originality: Training a diffusion model with a discriminative loss is not a very new concept. Please consider comparing with Discriminator Guidance [1] and Direct Discriminative Optimization [2]. They focused on real vs. fake, but real and fake can also be seen as a binary class.

- Poor baseline: The method only compares to the vanilla diffusion models. It could also be compared with the papers you cited as semi-supervised diffusion learning.

- Missing ablation on $\lambda$. What happens if $\lambda=0$?

- For SL, the author uses  $\lambda=1$, which is identical to the baseline. Then what is the meaning of Table 1?




[1] (ICML 23) Refining Generative Process with Discriminator Guidance in Score-based Diffusion Models


[2] (ICML 25) Direct Discriminative Optimization: Your Likelihood-Based Visual Generative Model is Secretly a GAN Discriminator

**Questions:**

- Without regularization in Eq.10, why does the denominator in Eq.9 diverge?

---

### Official Review · Reviewer_FDYt · 2025-10-29

**Soundness:** 1
**Presentation:** 2
**Contribution:** 1
**Rating:** 2
**Confidence:** 5

**Summary:**

This paper proposes reformulating the conditional diffusion model objective as a classification task. This approach allows weakly-supervised learning methods based on cross-entropy loss, originally developed for traditional classification tasks, to be directly applied to diffusion model training without requiring additional modifications. The paper presents examples focusing on two types of weak supervision: learning with noisy labels and learning from label proportions. Through toy experiments and evaluations on the MNIST and CIFAR-10 datasets, the paper demonstrates the applicability of the proposed method.

**Strengths:**

* Weakly-supervised learning is an important problem in diffusion models, yet it has been less explored compared to supervised learning tasks.
* In particular, diffusion models introduce the additional dimension of diffusion timesteps, which makes their application less intuitive compared to other generative models. Developing a unified framework to integrate this aspect can be highly valuable.
* The derivation of the proposed method is intuitive and well-written.

**Weaknesses:**

* In the Introduction, the motivation for reformulating diffusion models with a classification objective should be clarified. Simply stating that the "reverse direction" has been underexplored is not sufficiently strong reason. The fact that this reformulation enables the direct application of existing weakly-supervised learning methods based on cross-entropy loss could serve as a stronger motivation and should be mentioned early in the introduction.

* A theoretical justification is needed for why the classifier can be formulated as in Eq. (9). For example, it is known that the standard diffusion loss $l_{DM}$, based on the L2 loss, can be interpreted as a bound on the log-likelihood given an proper temporal weighting function (Song et al., NeurIPS 2021). In this paper, $p_\theta(y|x_t, t)$ is defined bia a softmax over $-l_{DM}$, and the model is trained using cross entropy $L_{ce}$. It remains unclear in which direction $p_\theta$ is optimized with respect to the data distribution under this formulation.

(Song et al., NeurIPS 2021) Maximum Likelihood Training of Score-Based Diffusion Models.

* The claim that the generative objective is not directly applied is somewhat questionable. Similar to the derivation of Appendix A, for general $\lambda_{reg}$, one can derive that $L_{CLDDM}$ is equivalent to $l_{DM}(x_t, t, y^*)+ (1-\lambda_{reg}) \log \sum \exp (- l_{DM})$. This indicates that the proposed objective is essentially a regularized version of the standard generative loss, where the additional term acts as a regularizer weighted by $(1-\lambda_{reg})$. Thus, I think that the main methodological difference lies in the introduction of $L_{reg}$.

* Given this, the role of $L_{reg}$ should be discussed beyond its function of preventing divergence in the denominator. Specifically, how does introducing $L_{reg}$ alter the training dynamics compared to using only $L_{ce}$? This discussion is currently missing.

* The current manuscript also appears to overclaim that equivalence between the CE-based formulation and the original generative objective. The equivalence only holds when the regularization term is included and $\lambda_{reg}=1$. Since the regularization term significantly affects the relationship between the two formulations, it should not be treated as a minor regularization for divergence control, but as a core component influencing equivalence.

* In the experiments, the supervised learning comparison is conducted with $\lambda_{reg}=1$, which makes the objective identical to the original generative formulation and thus trivially yields the same results. The paper should report and discuss the performance variations with respect to $\lambda_{reg}$, including results for different values in the supervised setting.

* Although the regularization term is described as "simple", in practice it requires computing $\epsilon$ for all conditions, which can be computationally expensive. The potential computational burden and scalability of the proposed method should be discussed. While the paper mentions using a multi-head setup from prior work, this approach may also introduce challenges. For example, the need to train new heads from scratch, which can be problematic for large diffusion models where pre-trained weights are typically required. Discussion on whether and how pre-trained model can be effectively utilized is needed.

* The method is only applicable to cross-entropy-style weakly supervised learning approaches. It would be better to discuss whether this limitation affects the generality of the framework. Moreover, the paper only experiments with relatively old and base WSL methods, which makes the empirical evaluation incomplete. So, the applicability to more recent WSL methods should be verified, as many of them operate under weaker assumptions and achieve improved performance compared to traditional methods.

* A comparison with existing diffusion-based WSL appraoches is necessary, both methodologically and experimentally. For example, it would be valuable to compare with Na et al. (2024), which also addresses learning under noisy labels, as mentioned in the paper, to better position the proposed method and demonstrate its advantages or differences.

* It would be beneficial to extend the experimental validation to higher-resolution datasets or more complex tasks (e.g., continuous or text-conditioned labels) to verify the method's scalability and general applicability.

* Minor points
  - In Eq. (4), the variance term on the right-hand side should be expressed in matrix form, i.e., $\beta_t I$.
  - In Eq. (10), the variable $y$ in $L_{ce}$ should be replaced with $y^*$.
  - In Section 4.1, the citation for Chen et al. (2024) should be corrected to: Chen et al., ICML 2024, Robust Classification via a Single Diffusion Model.

**Questions:**

Please provide explanations or clarifications to the points raised in the Weaknesses section.

---

### Official Review · Reviewer_ruds · 2025-11-01

**Soundness:** 2
**Presentation:** 2
**Contribution:** 2
**Rating:** 4
**Confidence:** 3

**Summary:**

The authors propose a Classifier-Driven Diffusion Model (CLDDM), which trains a diffusion model using a classification objective while still enabling data generation. They also extend the approach to weakly-supervised generation by leveraging established weakly-supervised learning techniques. Experiments are conducted on synthetic 2D Gaussian datasets and on image synthesis tasks using MNIST and CIFAR-10.

**Strengths:**

Elegant theoretical link between classification and diffusion training objectives.

Practical and flexible framework that supports weakly-supervised generation without requiring architectural changes.

**Weaknesses:**

Experiments are limited to low-resolution datasets (MNIST, CIFAR-10) and a toy 2D Gaussian example.

Analysis of scalability and performance on more complex or higher-resolution data is missing; only one network architecture is tested for images and a simple MLP for 2D Gaussian data.

No comparison with other related works.

The results between the standard diffusion model (DM) and CLDDM are very similar, making it harder to assess the advantages of the proposed method.

**Questions:**

How does the model scale to higher-resolution images (ImageNet? or even STL) or more complex datasets?

Can the CLDDM framework support more diverse weakly-supervised settings, such as partial or complementary labels, beyond noisy labels and label proportions?

How sensitive is the method to the choice of network architecture or hyperparameters?

Can the authors provide a quantitative analysis of training efficiency or sampling speed compared to standard diffusion models? does it converge faster?

---

### Note · Authors · 2025-11-17

**Comment:**

Thank you for the constructive reviews. We take the reviewer's suggestions into account and realize that the study needs improvement and the paper requires significant changes.

**Withdrawal Confirmation:**

I have read and agree with the venue's withdrawal policy on behalf of myself and my co-authors.